

# Global Distribution of Winter Lightning: a threat to wind turbines and aircraft

J. Montanyà[1], F. Fabró[1], O. van der Velde[1], V. March[2], E. R. Williams[3], N. Pineda[4], D. Romero[1], G. Solà[1]
and M. Freijo[1]

[1]Department of Electrical Engineering, Universitat Politècnica de Catalunya, Terrassa (Barcelona), 08222, Spain
[2]Gamesa Innovation & Technology, Sarriguren, (Navarra), Spain
[3]Massachusetts Institute of Technology, Cambridge, MA, USA
[4]Meteorological Service of Catalonia, Barcelona, Spain

*Correspondence to*: J. Montanyà (montanya@ee.upc.edu)

**Abstract.** Lightning is one of the major threats to multi-megawatt wind turbines and a concern for new generation of aircraft. That is because the use of lightweight of some composite materials. Both wind turbines and aircraft can initiate lightning and very favourable conditions for lightning initiation occur in winter thunderstorms. Moreover, winter thunderstorms are characterized for producing very energetic lightning. The paper reviews the different type of lightning interactions and summarizes the well-known winter thunderstorm areas. However, up to now comprehensive maps of global distribution of winter lightning prevalence to be used for risk assessment have been unavailable. In this paper we present the global winter lightning activity for a period of 5 years. Using lightning location data and meteorological re-analysis data, three maps are created: annual winter lightning stroke density, seasonal variation of the winter lightning and the annual number of winter thunderstorm days. In the northern hemisphere, the maps confirmed Japan to be one of the most active regions but other areas such as the Mediterranean and the US are active as well. In the southern hemisphere, Argentina and New Zealand experience the highest activity. The maps provided here can be used for providing risk assessment.

## 1 Introduction

Storms and lightning differ from one geographic area to another. In Europe, lightning activity is concentrated during the "warm season" since it is related to solar heating and availability of atmospheric water vapour (e.g. Poelman et al.,2015; Anderson and D. Klugmann, 2013). Poleman et al. (2015) found that winter months account only for 3 % of the annual lightning in Europe. But, although globally lightning activity associated to winter thunderstorms is low, these storms can produce very energetic lightning events and a large amount of damage (e.g. Yokoyama et al., 2014). Moreover, winter storms present the most favourable conditions for the initiation of upward lightning flashes from sensitive tall structures such as wind turbines (e.g. Montanyà et al., 2014) and for flying aircraft (e.g. Wilkinson et al., 2013). A recent study by Honjo (2014) of a sample of 506 lightning currents to wind turbines in Japan concludes that winter lightning currents tend to feature longer duration



currents, often bipolar, and that some particular wind turbines can be struck by lightning repeatedly in short periods of time. From the data, in about 5 % of the cases the charge transferred by lightning exceeded 300 C. Wang and Takagi (2011) analysed a sample of 100 records and summarized that 67.6 % of the cases presented negative polarity, 5.9 % presented positive polarity and 26.5 % presented bipolar currents. In that study they also found that about a 50 % of the cases were self-initiated by the

wind turbine and approximately the same percentage of flashes were initiated by other lightning activity. The authors noted that active thunderstorms produced more induced lightning than those storms with lower lightning activity. Additionally Wang and Takagi (2011) noted that strong wind conditions common in winter storms may favour upward lightning initiation.

Regarding aircraft, Murooka (1992) showed how lightning strikes to airplanes typically occur at lower altitudes during winter compared to summer. Gough et al. (2009) identified that 40 % of the studied lightning events to airplanes occurred during the

"cold season" which is not the period of the most frequent thunderstorm activity. Moreover, Wilkinson et al., (2009) concluded that because the lightning strike rate to helicopters at the North Sea during winter was much higher than expected, the presence of a helicopter actually triggers lightning. This phenomena present a significant safety risk to helicopters doing operations under this conditions.

The main goal of this paper is to present a global map of winter lightning. First, we summarize the lightning interactions

highlighting the situations related to winter storms. That is an important aspect which can vary according to the climatology of the thunderstorms for a particular area. Second, a global overview of winter storms is presented as well the resulting lightning maps that have been computed from global lightning data. These maps will provide a tool to identify risk areas of winter lightning when performing risk assessment.

## 2 Lightning interactions

Risk assessment can only be done effectively with a complete understanding of the interactions between lightning and the struck object. Wind turbines are tall structures and in this way can receive both downward and initiate upward lightning. In the case of an aircraft lightning is initiated in a bidirectional way (positive and negative leaders) from itself when flying in or beneath a thunderstorm. In this section we review the types of lightning related to wind turbines and aircraft. We do that because the mechanisms of lightning interaction with wind turbines and aircraft can differ from winter thunderstorms to

summer storms (storms associated with deeper convection).

First of all, downward lightning to wind turbines (Fig. 1a) can be more common in relation to deep convective situations (e.g. summer storms in the northern hemisphere and tropical storms). Downward lightning is the most frequent type of lightning and is also a threat to wind turbines and aircraft. The number of downward lightning events to a particular wind turbine will depend on the exposure of the turbine and the regional ground flash density.

In the case of upward lightning two situations are distinguished regarding the triggering of an upward leader from the turbine: self-initiated (Fig. 1d) and induced (Fig. 1b and 1c). We use the term induced lightning when it is related to the occurrence of another lightning flash which does not strike the turbine. In the case of induced upward flashes, a nearby CG flash or an IC



flash can provide the conditions for the inception of an upward leader. Upward induced lightning is more likely to occur during warm season storms because the high occurrence of lightning. Fig. 1b and 1c depict that situation. Under a thunderstorm, due to the intense electric fields produced by cloud charges, wind turbines can produce corona discharges. By means of corona, electric space charge is produced (positive for a typical dipole or tripolar charge structure as discussed Williams, 1989). As indicated by Montanyà et al. (2014a) this space charge screens the electric field at the turbine tip thereby preventing the initiation of a leader. In order to produce a stable leader, the field needs to increase at the tip of turbine (Bazelyan and Raizer, 1998). This increase of the electric field can be produced thanks to the fast neutralization of charge produced in a CG (e.g. Warner et al., 2012 and Montanyà et al., 2014b) or an IC flash. Because of the slow ion mobility of the space charge at the tip of the turbine, the electric field is not screened and it is increased. In the case of wind turbines, the most favorable conditions for induced triggered lightning will be the case of a fast and large charge neutralization in nearby CG and IC flashes and with enhanced electric fields due to the terrain height (close to the cloud charge) and orography (e.g. on mountain peaks).

A more favourable situation for self-initiated upward lightning is present in winter thunderstorms (Fig 1d). Resulting from the dependence of the electrification processes on temperature (e.g. Takahashi, 1984 and Saunders et al., 2006), in winter the cloud charges are located closer to the ground. But, even with the lower height of the cloud charges, winter storms are not prolific generators of downward lightning (Michimoto, 1993; López et al., 2012, Bech et al., 2013 and Hunter et al., 2001). That might be explained because of the necessity of opposite polarity charge under the main negative charge region necessary to initiate a leader in the cloud (Krehbiel et al., 2008). In the case of winter storms, Montanyà et al. (2007) showed that, because the low altitude of the freezing level (even at ground), the lower positive charge center in the cloud might not be accumulated and then downward lightning may not be initiated. But prominent objects on the ground or at mountain tops have favourable conditions to initiate an upward leader.

Another special situation characterized by energetic lightning is produced in relation to the stratiform regions of Mesoscale Convective Systems (MCS) (Fig. 1e). MCS are also common in winter storm structures (e.g. Mediterranean storms). MCS can present a higher percentage of positive CG lightning activity and higher peak currents than produced by cellular summer storms (e.g. MacGorman and Morgenstern, 1998). It is well known that intense +CG flashes occur in the stratiform regions of MCSs which also excite sprites in the mesosphere (e.g. Lyons, 1996; Williams et al., 2010; van der Velde et al., 2010 and Montanyà et al., 2011). These intense positive CG flashes can transfer hundreds of Coulombs of charge with continuing currents lasting up to tens of milliseconds (e.g. Li et al., 2008) which makes them a major threat to wind turbine and aircraft. However there is significant variation from one MCS to another.

Since most of the winter lightning strikes to turbines belongs to the upward lightning type (Honjo, 2015), the effect of rotation on the enhancement of lightning inception raised the interest (e.g. Rachidi et al., 2008, Wang et al, 2008, Montanyà et al., 2014a and Radicevic et al., 2012). However, there is no clear evidence that the number of lightning flashes increases significantly with the effect of rotation. In the studies by Wang et al. (2008) and Wang and Takagi (2011) in Japan the authors noted slightly larger number of strikes to rotating wind turbines than to a nearby protecting tall tower. Recently Montanyà et al. (2008) showed corona/leader activity associated with rotating wind turbines (Fig. 1 f). This activity can last for more than



an hour especially when the turbines are under an electrically charged stratiform region. This activity, even if it may not result in a complete lightning flash, can stress the dielectric properties of blades and needs to be considered in lightning protection standards.

In the case of aircraft, Fig. 1b would correspond to those cases of encounters between aircraft and lightning or the cases of
lightning initiated by the aircraft. There is no information about the occurrence of lightning that is initiated by the aircraft but induced by another lightning flash (Fig 1b and 1c). Initiation by aircraft can be more efficient when thundercloud charges are closer to the ground and charge may be larger because of less frequent discharging by lightning. That is the case of winter thunderstorms (Fig. 1d) and also the conditions under stratiform regions of MCS (Fig. e and d.). Regarding the situation in Fig. 1e also happens to aircraft where continuous corona discharges without resulting in lightning occurs. Montanyà et al.
(2014) showed a example.

## 3 Winter thunderstorms and global winter lighnting

### 3.1 Meteorology of winter thunderstorm areas

Thunderstorms develop as convective clouds to altitudes where it is cold enough for graupel and ice crystals to form and separate, creating layers of opposite cloud charges. Their development depends on the presence of conditional stability, with
temperatures decreasing with height over a large depth of the troposphere (steep lapse rates) while the boundary layer must contain sufficient amounts of water vapor whose latent energy is released in ascending parcels, causing positive buoyancy (see Wallace and Hobbs, 2006). In summer, water vapor is supplied by large evapotranspiration while diurnal heating is strong enough over land to create the needed vertical temperature gradients, often with help from the dynamics within low pressure systems. Convergence and ascending air near the surface is required to carry parcels to their level of free convection.
In the winter period, diurnal heating is weak and moisture content is much reduced over land. Low pressure systems are more vigorous and bring cold airmasses from polar and arctic regions southward to mid-latitudes, creating strong vertical temperature gradients as cold continental air flows over relatively warm seas and ocean currents. Air parcels near the surface then experience no inhibiting warm layers on their way to the equilibrium level, often the tropopause, and occur over large regions over sea behind cold fronts. The tropopause is found between 10-15 km at mid-latitudes in summer, but can descend
to 5-10 km in winter, limiting the vertical extent of convection. Just as in summer, low-level convergent winds organize the triggering of storms, but over sea these are often found near upwind coastlines, where enhanced friction and sloping terrain creates a relative stagnation and ascending flow.



### 3.1.1 Japan

In the northern hemisphere a well-known area of winter storms is found in Japan. There, three type of winter thunderstorms are identified in Adachi et al. (2005): thunderstorms associated with cold fronts crossing the Sea of Japan; thunderstorms systems originating in low-pressure areas over the Pacific Ocean; and thunderstorms originating in the Japan Sea Polar Air
Mass Convergence Zone (JPCZ). The storms originating in the JPCZ are due to advection of dry and cold air masses from the Eurasian continent. The interaction of these cold and dry air masses over the Sea of Japan leads to increases in the water vapor content by evaporation. Arriving on the Japan mainland, convergence with horizontal winds due to topographic effects produces strong updrafts supporting the formation of thunderstorms. Winter storms in Japan are known for the high occurrence of positive lightning compared to negative and bipolar lightning (e.g. Wu et al., 2014). This situation results in energetic
lightning in terms of total charge transfer and the numerous damages to wind turbines (Yokoyama et al., 2014 and Honjo, 2015).

### 3.1.2 Europe

In Europe, the prevailing Icelandic low and the Azores high pressure systems can produce intense low-pressure systems developing over the warm Gulf Stream. The largest amplitude systems transport cold unstable arctic airmasses into Western
Europe at their rear side and often stagnate in the area of the Mediterranean Sea, forming Genoa lows with a high activity of fall/winter thunderstorms. In mid winter, stable cold continental airmasses over Central Europe via the Balkans and France may also slide into the Mediterranean and produce shallow winter thunderstorms over the prevalent warm water there.

### 3.1.3 North America

In the United States, large winter depressions develop over the Gulf Stream and move north along the eastern coast (e.g. Dirks
et al., 1988) before reaching Canada. The energy to feed these storms originates in the air-sea interaction from the warm Gulf Stream water and baroclinic instability within the cold airmass over the continent. Blizzards form over the northeastern US and thunderstorms are produced as the cold front and cold continental airmasses flow out over the Great Lakes (with lake effect snow) and the warm Gulf Stream, where the cold front collides with warm subtropical air, producing linear thunderstorm systems. The west coast of the continent experiences similar conditions as western Europe, with cold unstable maritime
airmasses reaching mainly the shores of western Canada were lifting by the Rocky Mountains can initiate electrified convection.

Another cause of winter storms in Canada and the central US are outbreaks of artic fronts (e.g. Holle and Watson, 1992) in which cold artic air masses move from north to south, meeting warm humid air from the Gulf of Mexico. Along these cold fronts thunderstorms are formed in the warm elevated layer producing frozen precipitation at the surface.





### 3.1.4 General effect of ocean gyres

Every major ocean basin (North Atlantic, North Pacific, South Atlantic, South Pacific etc) contains a basin-scale rotating current flow—a gyre—with clockwise (counterclockwise) rotation in the northern (southern) hemisphere (Fig. 2). The primary drive for these gyres is the zonal wind stress from prevailing easterly winds in the tropics—otherwise known as the trade winds. In this near-equatorial portion of the gyre, the ocean surface is warmed substantially by sunlight and at the western limit of this equatorial transit, this warm oceanic flow is diverted northward and southward, depending on hemisphere. Since the surface air over continents is increasingly colder away from the equator and tends to be moving eastward off the continents at mid-latitude, one has a consistent situation in all gyres that warm ocean water in this poleward current is found beneath colder air away from the equator. This configuration is inherently unstable and can produce vigorous atmospheric convection and thunderstorm activity. The Gulf Stream along the North American coast and the Kuroshio Current along the eastern coast of Asia (China, Japan, Korea and Russia) are prime examples in which lightning activity over warm ocean water is prevalent during winter. In contrast, the return current on the eastern boundaries of oceanic gyres, and moving equatorward, is colder than the air overlying it. This situation is stable against convection and lightning is absent. A prime example is the Eastern Pacific Ocean. Similar behavior is present in the gyres of the southern hemisphere.

### 3.2 Global maps of winter lightning

In this section we present maps of global winter thunderstorms that are useful for risk assessment. In order to process the maps we used a simple criterion to define winter lightning conditions. The criterion to classify a lightning stroke as winter lightning is if it occurred in meteorological conditions with temperatures equal or lower than 5°C at the 900 hPa level (about 1 km above mean sea level). Temperature data at this pressure level is obtained on a 1°×1° grid from ECMWF Re-Analysis (ERA-Interim). Global lightning data were provided by the World Wide Lightning Location Network (WWLLN) (Rodger et al., 2006). The period of analysis correspond to five years (2009-2013). The results are presented in three maps: average annual stroke density, seasonal variation of the lightning stroke density and average number of winter thunderstorms per year.

Fig. 3 displays the global winter lightning distribution. The maximum annual stroke density was found to be no higher than 0.8 strokes·km-2·year-1, in order to present a more clear map, Fig. 3 has been limited to 0.2 strokes·km-2·year-1.

The map in Fig. 3 clearly shows the previously discussed areas of winter lightning (Japan, east of US, Mediterranean) and other areas with wind farms such as northeastern Argentina, southeastern Australia and western New Zealand. Fig. 4 plots the seasonal variation of the global winter lightning activity as stroke density.

The average stroke densities shown in the maps in Fig. 3 and 4 are influenced by the detection efficiency of the WWLLN. As any long range VLF lightning location system, WWLLN has a detection efficiency for each location that changes during the hours of the day due to VLF wave propagation, and also during time due to network upgrades, sensitivity of the sensors and data processing methods. The estimated overall stroke detection efficiency of WWLLN is considered to be 11 % according to Hutchins et al., 2012, Abarca et al. (2010) and Rodger et al (2009). Moreover, the WWLLN makes no distinction between





cloud-to-ground flashes and intra-cloud flashes. Then, the results presented in Fig. 2 and 3 are relative to WWLLN detections and cannot be adopted as absolute values.

Probably the most useful metric for evaluation of the winter lightning risk is the number of winter thunderstorm days per year ($T_w$). Here $T_w$ is obtained in a 1°×1° grid. A winter thunderstorm day in a grid cell is counted if at least one lightning stroke

agreeing with the presented temperature-pressure level criterion is detected within the cell. The $T_w$ map is depicted in Fig. 5.

## 4 Discussion

Areas of winter lightning are well defined outside the ITCZ. The resulting maps show that contrary to what occurs with deep convective storms, winter lightning activity is more distributed over the oceans than over continental areas. One of the reasons is the role that warm oceanic water plays to produce energy for convection during cold seasons. In Fig. 2 we resumed the main

oceanic current gyres. Note how this simple picture goes a long way in explaining the patterns of winter lightning in Fig. 3 and 5. The preference for oceanic and coastal areas is an important aspect for coastal onshore and offshore wind farms.

Regarding winter thunderstorm activity and risk assessment, the maps indicate some particular areas more critical to winter lightning because of prevalence in larger areas onshore in proximity to active ocean areas. The Japan mainland is surrounded by winter thunderstorms, the map in Fig. 5 shows how especially the west coast is particularly active reaching in some areas

about 30 days of winter thunderstorms per year. Another particularly active region that affects extensive onshore areas is found in the Mediterranean Sea (e.g. South Italy and the Balkans) and in New Zealand. In the case of North America, the highest number of annual winter thunderstorm days is located in the Atlantic Ocean. Although the central eastern regions of the US the number of winter thunderstorms are not so high, the lightning stroke densities are significant. Other areas sensitive to winter thunderstorms because present installations and also ongoing offshore farms, range from the northern coast of Spain,

western coast of France, and the European coast in the North Sea. In addition, northeastern Argentina and southern Australia must be highlighted as well.

For risk assessment it is convenient to consider the Tw provided in Fig. 5 as the first indicator of risk to winter lightning activity. In addition to identification of winter thunderstorm areas, the seasonal variation of the winter lighting activity (Fig. 4) is another aspect deserving attention for planning purposes, for equipment maintenance or to setup lightning warnings (e.g.

European standard EN 50536 (2011). That is also important when working with tall cranes. In the northern hemisphere the activity is concentrated from October to June whereas in the southern hemisphere the activity if significant from April to September.

## 5 Conclusions

Winter lightning poses a critical risk to tall objects such as wind turbines and also to flying aircraft because these have

favourable conditions for self-lightning initiation. In this paper we presented for the first time world maps with winter lightning activity. The maps shows that winter lightning occurs in extratropical regions with preference for oceanic and coastal areas in the western limits of permanent oceanic gyres. As a general conclusion, winter lightning maps presented in this work suggest





that winter activity in Japan may be the highest, as supported by the previously discussed works. But Japan is not an exclusive region for winter lightning but other areas such as the Mediterranean and the US are active as well. In the southern hemisphere, Argentina and New Zealand experience the highest activity.

The maps are helpful for risk assessment analysis such as proposed in standards (e.g. IEC) providing a tool to identify areas of winter lightning activity. In these areas tall structures such as wind turbines can be submitted to very energetic lightning and to an environment favourable for self-lightning initiation. Risk assessment of the effect of winter lightning shall also include the exposure. In the case of wind turbines, those turbines located in areas influenced by winter lightning at high altitudes can experience very high number of lightning flashes. Also high risk locations are those offshore (e.g. offshore wind turbines, platforms and helicopter operations at those sites). In these situations the new presence of a tall object can significantly increase the winter lightning activity.

Finally, the simple methodology employed to classify a lightning stroke as being winter lightning will allow to conduct further risk assessment based on local lightning data and meteorological data.

*Acknowledgements.* This work was supported by research grants from the Spanish Ministry of Economy and Competitiveness (MINECO) AYA2011-29936-C05-04 and ESP2013- 48032-C5-3-R. This work has been part of the author's activity in the CIGRE WG C4.36 "Winter Lightning – Parameters and Engineering Consequences for Wind Turbines".

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





Figure 1. a) Downward lightning stroke to a wind turbine; b) Upward lightning initiated by a nearby CG flash; c) Upward lightning initiated by a IC flash; d) Upward lightning from a wind turbine and lightning initiated by aircraft in a winter storm (case of Japan); e) Lightning in MCS; f) Repetitive corona/leader emissions from wind turbines under storms. Proportions are not meet in the representations.



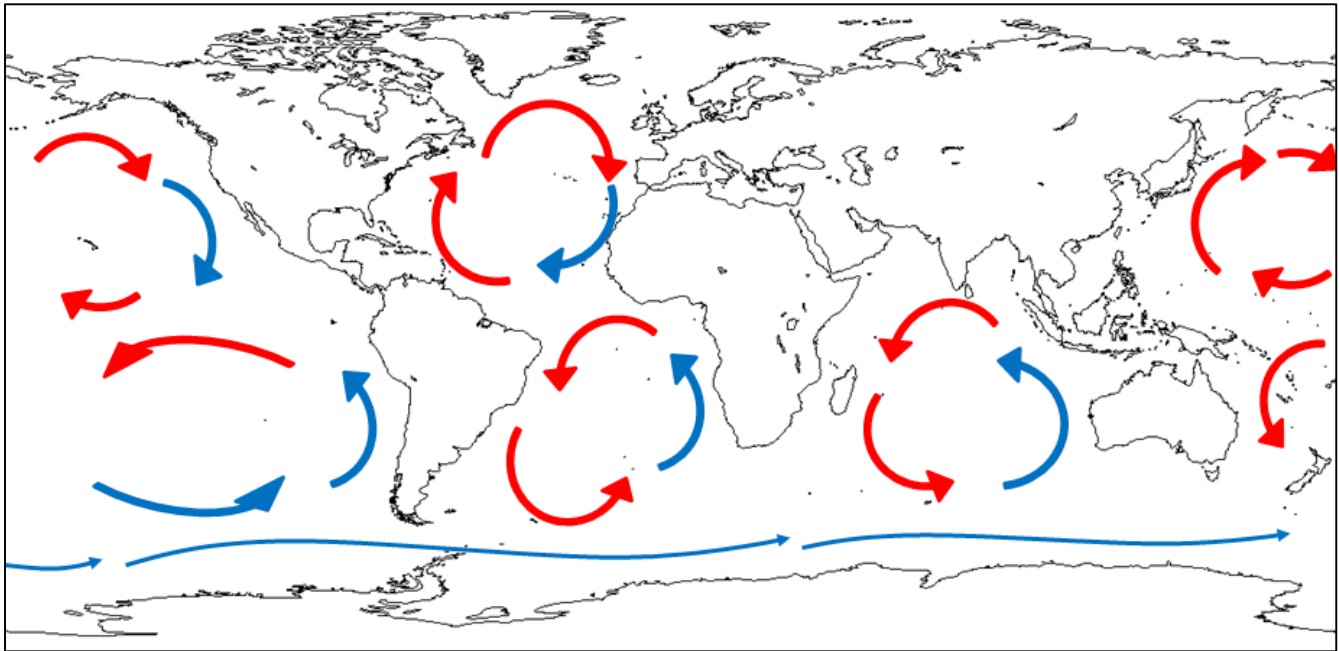

Figure 2. Simplified distribution of the most significant rotating oceanic currents: North Atlantic Gyre, South Atlantic Gyre, North Pacific Gyre, South Pacific Gyre, Indian Ocean Gyre and the Artic Circumpolar Gyre (for more information see Siedler

15   et al., 2013). Red and blue arrows indicate warm and cold currents, respectively.

20





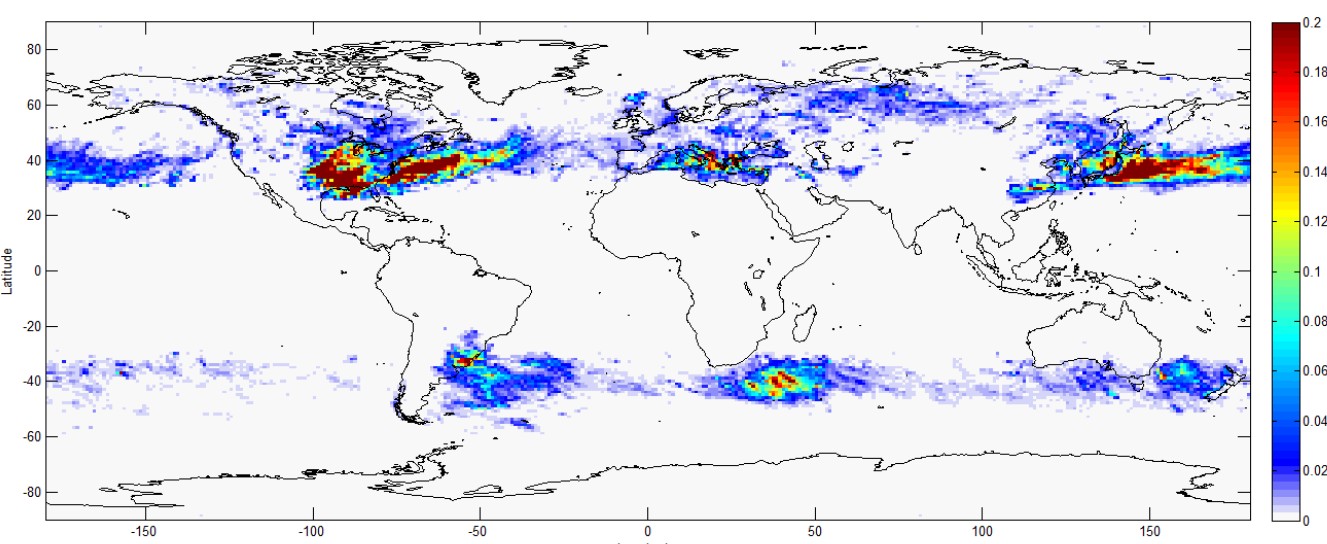

Figure 3. Global distribution of winter lightning stroke density (strokes·km-2·year-1) for the period 2009-2013.

20



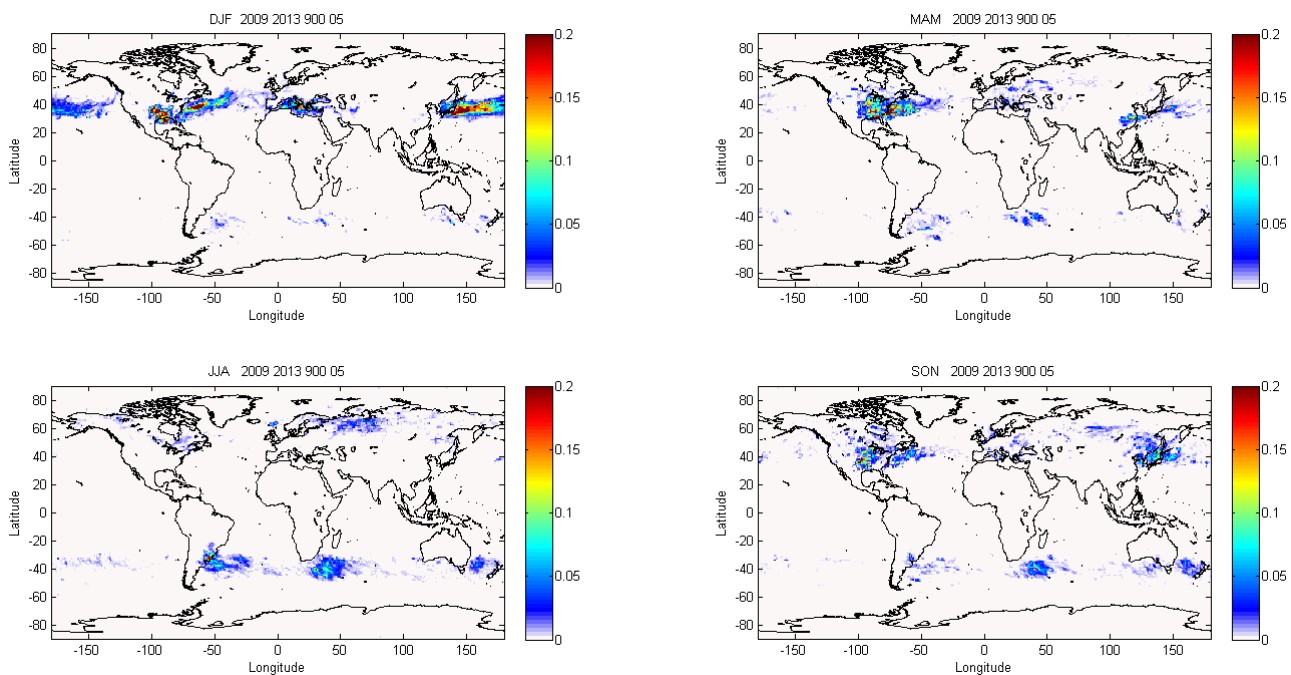

Figure 4. Seasonal variation of the winter lightning stroke density (strokes·km-2·year-1) distribution for the period of 2009-
10   2013. Note that major shift of activity from DJF and MAM periods in the northern hemisphere to more JJA and SON periods
in the southern hemisphere.



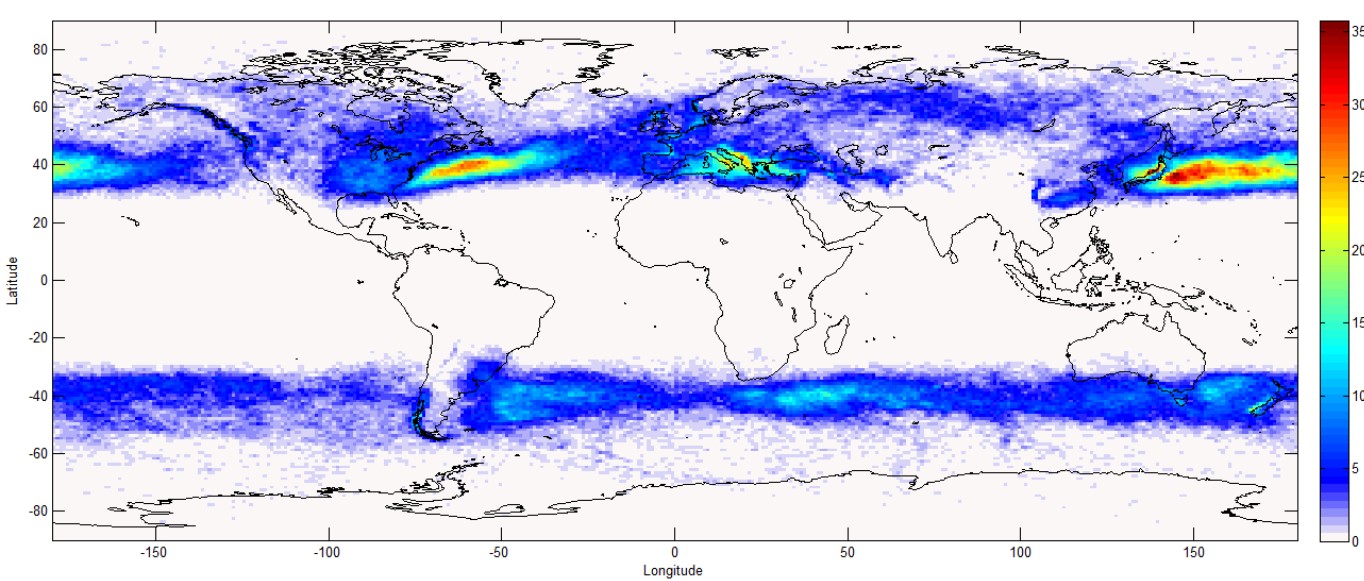

10 Figure 5. Average number of winter thunderstorms per year ($T_w$) for the period 2009-2013.

20