# Peer review of "Global Distribution of Winter Lightning: a threat to wind turbines and aircraft"

_Natural Hazards and Earth System Sciences, 2015_

## Referee Comment (RC1) · Anonymous Referee #1 · 21 Jan 2016

The paper is the first to discuss the global activity of winter-type lightning as the authors write, and is valuable to the community considering the importance of lightning protection against this type of lightning. It requires some revision, however, e.g. it lacks explanation on the definition of winter lightning, and an important aspect of winter lightning around Japan. It follows specific comments.

Fig. 1 d) does not match the explanation of page 3, 2nd paragraph, where is written that lower positive charge might not be accumulated.

Fig. 1 e) is wrong. Page 3, 3rd paragraph is explanation of downward positive flash (the cited references are for summer MCS). Fig. 1 e) has to be a cartoon of a downward positive flash.

page 3, line 27: This paragraph refers to downward positive flashes, which rarely strike

wind turbines or aircrafts in the cold season, too. Upward positive flashes are the threat to wind turbines in winter, and this is a different phenomenon from that explained in this paragraph.

page 5, Section 3.1.1: Adachi et al. examined sprite-producing storms only. Sugita and Matsui (2008) report on other type of lighting activity in the cold season. They call it "isolated type", producing only a few lightning flashes a day, and is clearly distinguished from JPCZ-type, which is much more active (may correspond to Sugita and Matsui's coastline type). The isolated type may not produce sprites, but contributes significantly to winter thunderstorm days of over 30 around Japan. reference: Akiko Sugita, Michihiro Matsui, "Examples of winter lightning observed by the JLDN", 2008 ILDC/ILMC, Tucson, Arizona, 2008. http://www.vaisala.com/en/events/ildcilmc/Documents/Examples%20of%20Winter%20Lightning%20Observed%20by%20th

page 5, Section 3.1.2 does not have a reference.

page 6, Section 3.2, 1st paragraph: Basis is not given to the criterion of winter lightning, "temperatures equal or lower than 5°C at the 900 hPa level". Saito et al. (2012) suggest a boundary of 5.7 km of -10ïĆřC level, based on their observation. reference: M. Saito, M. Ishii, F. Fujii, M. Matsui: Seasonal Variation of Frequency of High Current Lightning Discharges Observed by JLDN, IEEJ Trans. P&E, Vol. 132, No. 6, pp. 536-541, 2012.6.

page 6, Chapter 4 and Fig. 5: Tw should be "the number of winter thunderstorm days per year" and not the number of thunderstorms.
* * *

---

## Referee Comment (RC2) · Anonymous Referee #2 · 17 Feb 2016

The paper "Global Distribution of Winter Lightning: a threat to wind turbine and aircraft" provides new analysis of global lightning data, focusing on the distribution of winter thunderstorms, that are known to be threat to aviation and wind turbines. The data analysis approach is novel, as provides a useful comparison of the relative intensity of regions of winter thunderstorms. The quality of the analysis and report suggests that it is of a sufficient standard to be published without major corrections; however the number of minor corrections required is significant. A very high proportion of these are due to language issues, where the point made is acceptable, but the wording requires clarification. Additionally, there are also a number of scientific questions that need to be addressed.

- GENERAL COMMENTS

[Figure]

The use of a threshold of 5°C at 900hPa is not sufficiently explained/justified. Why isn't 4°C, or 6°C, appropriate? What is the impact of using different temperature/pressure levels? It may be that this approach has been justified in existing literature, but if this is the case, an appropriate reference would be required.

The use of these specific criteria also means that thunderstorms at high latitudes in the summer are still classified as "winter" thunderstorms. This can be seen in Figure 4, where lightning in the meteorological summer is classified as winter lightning in North America and northern Asia (bottom left subfigure) and near to the Southern Atlantic and Southern Indian Ocean (top right subfigure). The term is appropriate in general, but this characteristic should be discussed, as it may be that these (technically) summer thunderstorms actually exhibit characteristics similar to winter thunderstorms (i.e. in terms of the heights of charged regions).

Although the comparison of the relative intensity of global regions of winter lightning activity is useful, the authors do not provide enough analysis of the global variability in WWLLN detection efficiency (DE).In page 6, line 31, they state that the DE of WWLLN "is considered to be 11%". There will be significant variations in this globally, however, based on sensor distribution. Further to this, Rudlosky and Shea (2013) showed that WWLLN is apparently three times more likely to detect a flash over the ocean than over land, based on comparison with LIS data. (Rudlosky, Scott D., and Dustin T. Shea. "Evaluating WWLLN performance relative to TRMM/LIS." Geophysical Research Letters 40, no. 10 (2013): 2344-2348)

The figures present the results in units of strokes per square kilometer per year. Is there a reason that WWLLN fixes were not merged into flashes using suitable space/time criteria? This would make the data easier to compare with satellite data (OTD, LIS, or the upcoming GLM and MTG-LI) and other analyses of lightning density, which also generally use flashes.

The middle paragraph of the Discussion section (Page 7, lines 12-21) seems a little

muddled. The authors leap from a one-sentence analysis of one area, to another area, to another. The Mediterranean and New Zealand are mentioned before North America and the North Atlantic, despite the latter experiencing much higher winter lightning densities than the former, based on Figure 3. More context in the text, i.e. peak densities or winter thunderstorms days, would make this section simpler to understand without constantly needing to refer to the figures.

The colour scales used in Figures 4 and 5 could be adjusted to allow for the appreciation of greater detail in regions of stroke densities above 0.2 strokes per square kilometer per year. Large regions of North America, North Atlantic and the North Pacific are simply displayed as block red in Figure 3. The current colour scale between 0.0 and 0.2 could be kept the same, so that 0.2 is still red, but values about that could transition into black/grey/purple, for example. See Anderson and Klugmann (2014) as an example.

There are a large number of specific comments and technical corrections that are simply due to the use of language, but do not prohibit understanding of the topics discussed. References were generally accessible and appropriate for the text, unless otherwise stated in the specific comments.

- SPECIFIC COMMENTS

Page 1, line 9: ". . . for new generation of aircraft." does not read well in English. Suggest ". . . for modern aircraft."

Page 1, line 10: "That is because the use of lightweight of . . ." does not read well in English. Suggest merging with the previous sentence: ". . . for modern aircraft, due to the use of lightweight composite materials."

Page 1, line 14: ". . . characterized for producing . . ." does not read well in English. Suggest ". . . characterized by a relatively high proportion of . . ."

Page 1, line 14: ". . . type . . ." Should be ". . . types . . ."

Page 1, line 15: "However, up to now ..." The however is unnecessary, and "up to" does not read well. Suggest "Until now ..."

Page 1, line 17: "... three maps ..." Actually, six maps are presented, in three figures.

Page 1, line 21: "... provided here can be used for providing ..." Double use of forms of the word "provide" does not read well. Further, the information presented in the manuscript is not sufficient for an entire assessment of risk on its own. Suggest "... provided here can be used in the development of a ..."

Page 1, line 26: ""But, although ..." Word "But" is not necessary.

Page 1, line 26: "... winter thunderstorms is low ..." It would read better is this were put in context. Suggest "... winter thunderstorms is relatively low compared with summer thunderstorms ..."

Page 2, line 9: "... events to airplanes ..." does not read well in English. Suggest "... events involving airplanes ..." or "... events involving lightning connections to airplanes ..."

Page 2, line 12: "This phenomena present ..." Mixes singular and plural, should read "This phenomena presents ..."

Page 2, line 13: "... under this conditions." Mixes singular and plural, should read "... under these conditions."

Page 2, line 15: "... interactions highlighting the situations related ..." does not read well in English. Suggest "... interactions related ..."

Page 2, line 15: "That is ..." does not read well in English. Suggest "This is ..."

Page 2, line 21: "... can receive both downward and initiate upward ..." The word "both" refers to receiving downward and initiating upward lightning, and so should appear earlier. Suggest "... can both receive downward and initiate upward ..."

[Figure]

Page 2, line 23: "We do that . . ." does not read well in English. Suggest "We do this
. . ."

Page 2, line 31: Fig. 1d is referred to in the text before Figs 1b and 1c. It would make
sense then to either rearrange this sentence, so Fig. 1d is mentioned after the previous
figures, or to rearrange the subfigures in Figure 1 so that what is currently 1d becomes
1b, 1b becomes 1c and 1c becomes 1d.

Page 3, line 1: It has been contested whether IC lightning is technically a "flash" in the
same way as CG lightning, as there is not a stepped leader/return stroke structure. I
have debated this previously with other scientists, who prefer the term "IC discharge".
The change here is not an absolute necessity, however.

Page 3, line 1: "Upward induced lightning is more likely to occur . . ." It would be useful
to clarify whether this refers to absolute numbers of upward induced lightning events or
relative numbers. As an arbitrary example, if the relative proportion of events drops by
50%, but the total amount of lightning increases by a factor of 4, the absolute number
doubles.

Page 3, line 16: ". . . because of the necessity of . . ." This does not read well in English.
Suggest ". . . because the presence of . . ."

Page 3, line 16: ". . . charge region necessary to . . ." This does not read well in English.
Suggest ". . . charge region is necessary to . . ."

Page 3, line 29: ". . . turbines belongs to . . ." Mixes singular and plural, should read ". . .
turbines belong to . . ."

Page 3, line 30: ". . . raised the interest." This does not read well in English. Suggest
". . . has been discussed and investigated."

Page 4, line 1: ". . . an hour especially when . . ." Suggest inserting a comma, i.e. ". . .
an hour, especially when . . ."

Page 4, lines 9-10: "Montanyà et al. (2014) showed a example." This is not a full sentence. "a" should read "an". Suggest merging with the previous sentence, i.e. "... lightning occurs, as observed by Montanyà et al. ..."

Page 4, line 15: "... with temperatures decreasing with height over a large depth of the troposphere (steep lapse rates)". The atmosphere can be absolutely stable, even if the temperature decreases with height. This comment is not strictly necessary, so I suggest removing it and allowing the interested reader to check the associated reference for further detail on conditional stability.

Page 4, line 16: "... vapor whose latent ..." Water vapor is not a "who". Suggest "... water vapor from which latent ..."

Page 4, line 19: "Convergence and ascending air near the surface is ..." Mixes singular and plural, should read "Convergence and ascending air near the surface are ..."

Page 4, line 20: "... moisture content is much reduced ..." This is not absolutely always the case, only generally. Suggest "... moisture content is generally much reduced ..."

Page 4, line 20: "Low pressure systems are more vigorous ..." This is not absolutely always the case, only generally. Suggest "Low pressure systems are generally more vigorous ..."

Page 4, line 21: "... polar and arctic regions southward to mid-latitudes, ..." This section is talking globally, not only about the northern hemisphere. Suggest "... polar regions towards mid-latitudes, ..."

Page 4, line 23: "... then experience no inhibiting warm layers on their way to the equilibrium level ..." It is possible that warmer layers are encountered, but that are insufficiently warm to remove the buoyancy of the rising air parcel. Suggest "... then gain sufficient buoyancy to ascend large vertical distances ..."

Page 4, lines 23-24: "... and occur over large regions over sea ..." This implies air parcels only occur behind cold fronts. Suggest removing "and occur".

Page 4, line 27: ". . . a relative stagnation . . ." It is not clear what this means. Suggest replacing with ". . . convergence . . ."

Page 5, line 2: ". . . three type of . . ." Mixes singular and plural, should read ". . . three types of . . ."

Page 5, line 3-4: ". . . thunderstorms systems . . ." Mixes singular and plural, should read ". . . thunderstorm systems . . ."

Page 5, line 10: ". . . and the numerous damages to . . ." This does not read well in English. Suggest ". . . and a high rate of occurrence of damage to. . ."

Page 6, line 6: ". . . diverted northward and southward, depending on the hemisphere." It would help to be specific here. Suggest either ". . . diverted northward (southward) in the northern (southern) hemisphere" or ". . . diverted poleward".

Page 6, line 9: "This configuration is inherently unstable . . ." It would be better to say "This configuration drives the development of atmospheric instability . . ."

Page 6, line 16: ". . . that are useful . . ." This has not been conclusively demonstrated yet. Suggest ". . . that may be of use . . ."

Page 6, lines 28-29: "As any long. . ." This does not read well in English. Suggest "As with any long . . ." Page 6, line 31-32: ". . . 11% according to . . ." Remove the "according to", and the "and" later in this sentence, and enclose the references in parenthesis.

Page 7, line 3: "Probably the most useful metric. . ." The most useful metric would be data from a high detection efficiency network with well understood weaknesses covering the area of interest for an extended number of years. That way, the true CG flash rate, and thus the true risk, could be estimated to a high precision. Suggest "An improved metric. . ."

Page 7, line 9: ". . . we resumed the . . ." This does not make sense. Suggest ". . . we summarized the . . ."

Page 7, line 12: "... more critical to ..." This does not read well in English. Suggest "... more vulnerable to ..."

Page 7, lines 13-14: "The Japan mainland is surrounded by winter thunderstorms..." This sounds like the thunderstorms are permanent. Suggest "Waters around the Japanese mainland are particularly susceptible to winter thunderstorms..."

Page 7, lines 14-15: "... particularly active reaching in some areas about 30 days of winter thunderstorms ..." This does not read well in English. Suggest "... particularly active with some areas experiencing about 30 days of winter thunderstorms ..."

Page 7, lines 16-17: "In the case of North America, the highest number of annual winter thunderstorm days is located in the Atlantic Ocean." North America and the Atlantic Ocean are geographically separate, this sentence is not well phrased.

Page 7, line 17: "Although the central ..." Should include "in", i.e. "Although in the central ..."

Page 7, line 17-18: "... US the number of winter thunderstorms are not so high ..." This does not read well in English. Suggest "... US winter thunderstorms are not as frequent ..."

Page 7, line 22: "... as the first indicator of risk to winter ..." This does not read well in English. Suggest "... as an initial indication of the risk of winter ..."

Page 7, line 25: "That is also important ..." This does not read well in English. Suggest "Lightning risk is also important..."

Page 7, lines 25-27: It is stated in the last sentence of the Discussion that winter thunderstorm activity is "concentrated from October to June", i.e. in 75% of the year. I would view a concentration as generally referring to a minority of a sample.

Page 7, line 30: "... for self-lightning initiation." This does not read well in English. Suggest "... for lightning self-initiation."

Page 8, line 1: "But Japan is . . ." This does not read well in English. Suggest "Japan is. . ."

Page 8, line 2: ". . . lightning but other . . ." This does not read well in English. Suggest ". . .lightning, as . . ."

Page 8, line 4: "The maps are helpful for risk assessment . . ." This has not been conclusively demonstrated yet. Suggest "The maps may be of use for risk assessment . . ."

Page 8, line 4: "IEC". Expand this acronym.

Page 8, line 5: ". . .turbines can be submitted to . . ." This does not read well in English. Suggest ". . . turbines can be exposed to . . ."

Page 8, line 6 ". . . for self-lightning initiation." This does not read well in English. Suggest ". . . for lightning self-initiation."

Page 8, line 8: "Also high risk locations are those offshore. . ." This does not read well in English. Suggest "Locations at the greatest risk tend to be offshore. . ."

Page 8, line 9: ". . . the new presence of . . ." This does not read well in English. Suggest ". . . the installation of . . ."

Page 8, line 11: ". . . winter lightning will allow to conduct further . . .." This does not read well in English. Suggest ". . . winter lightning may be beneficial in conducting . . ."

Page 8, line 12: ". . . on local lightning data and meteorological data." This does not read well in English. Suggest ". . . on local combined lightning and meteorological data."

Page 9, lines 21-23: Montanyà el al. (2011) reference not generally available online.

Page 9, line 29: Murooka (1992) reference not accessible unless via web page in Japanese.

Page 10, line 21-22: Wang and Takagi (2011) not generally available online.

[Figure]

Figure 1: Subfigures 1a and 1b imply that the CG lightning is initiated from the small positively charged region at the base of the cloud. Assuming these figures represent negative CG lightning, it would be better if the branching of the lightning were to extend further into the negatively charged region, and were to spread out within that region (Montanya et al. 2014a, Figure 3 demonstrates the extent of a CG flash within a cloud nicely).

Figure 1, caption: Remove "(Case of Japan)", this does not make sense.

Figure 1, caption: "Proportions are not meet in the representations." This does not read well in English. Suggest "Proportions in these diagrams are not to scale."

Figures 3-5: The font within the figures is slightly too small, and difficult to read. The coastlines and edges of the plots are also very fine. It may be that the figures were created at one size, then had to be scaled down to fit the page. It would be preferable to scale down the original image, so that when it is reproduced in print, the text and edge lines appear less fine.

Figure 4: I have a suspicion that the values of the flash densities in this figure are wrong. If flash densities across North America/Japan are well in excess of 0.2 flashes per square kilometer per year annually, and there are seasons where the values are well below this level, there must be seasons where the flash density is significantly greater than the annual average. Looking at the peak densities in Figure 4, this is not the case. What I suspect has happened is that the number of flashes per grid box have only been divided by the area, to give units of flashes per square kilometer per season. In fact they must then be divided by the number of days in the year and multiplied by the number of days in the season to convert the units to flashes per square kilometer per year. This means that all of the densities are too low by a factor of four.

Figure 4: Subfigure titles contain "900 05": Presumably these are the temperature height and cut-off settings for the data. These would preferably be removed.

Figure 4: The details in the maps are hard to see in this plot. There are two options that would improve this situation. One would be to use a single large, narrow colorbar to the right of all four figures, as the same scale is used in each, and the amount of whitespace could also be reduced, to make better use of the available space. Alternatively, the maps could be rotated by 90 degrees to fill a page sideways, which would allow for a lot more detail to be visible.

- TECHNICAL CORRECTIONS

Page 1, line 24: "et al.,2015" should include a space after of comma, i.e. "et al., 2015".

Page 1, line 24: "Anderson and D. Klugmann" should not include initial, i.e. "Anderson and Klugmann".

Page 1, line 25: "Poleman" should be spelt "Poelman".

Page 1, line 29: "(Montanyà et al., 2014)". There are two Montanyà et al papers in 2014, this reference presumably refers to Montanyà et al., 2014a?

Page 1, line 29: "Honjo, 2014". Wrong year, should read "Honjo, 2015".

Page 2, line 10: "...Wilkinson et al., (2009) concluded..." Comma should be removed.

Page 2, line 10: "...Wilkinson et al., (2009) concluded..." Incorrect year: should be 2013.

Page 3, line 31: Missing caron and acute from name of author: Radicevic should read "Radičević".

Page 3, line 33: "Recently Montanyà et al. (2008) ..." No such reference: suggest Montanyà et al. (2014a).

Page 3, line 34: "Fig. 1 f". Space between figure number and subfigure letter should be removed.

Page 4, line 4: "Fig. 1b" should refer to "Fig. 1d".

Page 4, line 8: "Fig. e and d" should refer to "Fig. 1e and 1f".

Page 4, lines 9-10: "Montanyà et al. (2014)". There are two Montanyà et al papers in 2014, this reference presumably refers to Montanyà et al. (2014a)?

Page 4, line 11: "Lightning" misspelled as "Lighnting".

Page 5, line 16: "mid winter" should be "mid-winter".

Page 5, line 25: "were" should be "where".

Page 5, line 27: "artic" should be "Arctic".

Page 5, line 27: "Holle and Watson, 1992". Wrong year, should read "Holle and Watson, 1996".

Page 7, line 1: "Fig. 2 and 3" should be "Fig. 3 and 4".

Page 7, line 25: Missing closing parenthesis.

Page 8, line 19: Page numbers not needed in Abarca et al. (2010) reference.

Page 8, line 22: Reference requires full title of paper: "A European lightning density analysis using 5 years of ATDnet data"

Page 8, line 23: Incorrect year in Anderson and Klugmann (2013) reference: should be 2014.

Page 8, line 26: Missing "and" between second to last and last author.

Page 8, line 33: Incorrect year in Holle and Watson (1992) reference: should be 1996.

Page 9, line 6: Page numbers not needed in Hutchins et al. (2012) reference.

Page 9, line 25: Missing space in journal title: "Res.Atmos." should read "Res. Atmos."

Page 9, line 30: Reference requires full title of paper: "The European lightning location system EUCLID – Part 2: Observations"

Page 10, line 1: Missing caron and acute from name of author: Radicevic should read "Radičević".

Page 10, line 1: Missing space after colon: "Badea, I:Impact" should be "Badea, I: Impact"

Page 10, line 10: Incorrect page range: "2653-2673" should be "2653-2674".

Page 10, line 29: Missing space and full stop in author's name: "Williams, E.R," should be "Williams, E.R.,"

Page 10, line 34: Missing indent at start of line.

Figure 2: Caption reads "artic", should read "Antarctic".

---

## Author Comment (AC1) · 8 Mar 2016

The authors thank very much the reviewers for their useful comments and remarks. First, a lot of corrections are made to improve the manuscript. All remarks of the reviewers are considered and some necessary changes are made in the sense of these remarks. For example: the definition of winter lightning is included and the criterion of classification revised according to the suggestions of the reviewers.

Below we include the responses to each of all referee's comments:

- Responses to Referee' #1 comments are found from page 1 to page 3.
- Responses to Referee' #2 are found from page 4 to page 14.

After page 14 we include the manuscript with tracking changes.

**The paper is the first to discuss the global activity of winter-type lightning as the authors write, and is valuable to the community considering the importance of lightning protection against this type of lightning. It requires some revision, however, e.g. it lacks explanation on the definition of winter lightning, and an important aspect of winter lightning around Japan. It follows specific comments.**

**Fig. 1 d) does not match the explanation of page 3, 2nd paragraph, where is written that lower positive charge might not be accumulated.**
Agree, the figure 1.d) shows low positive charge. It is more consistent to refer in terms of the necessity of charge of opposite polarity below the mid-level charge in order to favor the production of CG. The text is clarified. It is indicated that the reference of Montanyà et al., 2007 is for European winter storms and the case in figure 1d it is already mentioned that the charge structure belongs to the case of Japan.

**Fig. 1 e) is wrong. Page 3, 3rd paragraph is explanation of downward positive flash (the cited references are for summer MCS). Fig. 1 e) has to be a cartoon of a downward positive flash.**
Thanks for the comment. In figure 1e we only include upward lightning from wind turbines which can be positive flash (upward negative leader). The figure has improved adding a downward positive flash. Figure caption is clarified indicating that both positive upward and downward flashes occurs at the stratirform region of the MCS.

**page 3, line 27: This paragraph refers to downward positive flashes, which rarely strike wind turbines or aircrafts in the cold season, too. Upward positive flashes are the threat to wind turbines in winter, and this is a different phenomenon from that explained in this paragraph.**

Agreed, the text is clarified according to the comment. We also indicate that downward positive flashes striking close a tall structure can initiate an upward flash.

**page 5, Section 3.1.1: Adachi et al. examined sprite-producing storms only. Sugita and Matsui (2008) report on other type of lighting activity in the cold season. They call it "isolated type", producing only a few lightning flashes a day, and is clearly distinguished from JPCZ-type, which is much more active (may correspond to Sugita and Matsui's coastline type). The isolated type may not produce sprites, but contributes significantly to winter thunderstorm days of over 30 around Japan. reference: Akiko Sugita, Michihiro Matsui, "Examples of winter lightning observed by the JLDN", 2008 ILDC/ILMC, Tucson, Arizona, 2008. http://www.vaisala.com/en/events/ildcilmc/Documents/Examples%20of%20Winter%20Lightning%20Observed%20by%20the%20JLDN.PDF**
Thanks so much for the reference. We've included it and add the isolated type to the section.

**page 5, Section 3.1.2 does not have a reference.**
There is no a particular reference dealing specifically with winter thunderstorms in Europe in which the particular meteorology of these storms is described. Perhaps the best descriptions can be found in Estofex (http://www.estofex.org/). We included that reference as well Holley et al. 2014 for northern Europe winter storms.

**page 6, Section 3.2, 1st paragraph: Basis is not given to the criterion of winter lightning, "temperatures equal or lower than 5 C at the 900 hPa level". Saito et al. (2012) suggest a boundary of 5.7 km of -10 C level, based on their observation. reference: M. Saito, M. Ishii, F. Fujii, M. Matsui: Seasonal Variation of Frequency of High Current Lightning Discharges Observed by JLDN, IEEJ Trans. P&E, Vol. 132, No. 6, pp. 536-541,b2012.6.**

Thanks for the reference. Actually Saito et al., 2012 shows an average temperature of -10 ºC at 2.7 km (~700 hPa) for the classified thunderstorms as winter. We observed the same in Montanyà et al. (2007) in Europe and a recent publication by Warner et al. (2014) for the US.

It is difficult to establish a precise criterion and easy enough to allow the global classification of winter lightning and thunderstorms. Our criterion of temperatures equal or lower than 5 ºC at 900 hPa produce good agreement with the observations indicated in the referenced papers. But we decided to modify the criterion to (700 hPa and -10ºC ) in the way to be more consistent with the references of Saito et al. (2012), Montanyà et al. (2007) and Warner et al. (2014). We choose actually, our criterion of winter lightning to be those lightning flashes occurring in cold d airmass thunderstorms. Then a criterion of -10 ºC at 700 hPa is more reasonable in order to identify airmass thunderstorms and not include those storms such as winter squall lines (common in the US) which are associated with warm and stable air above the freezing air. The results obtained are very similar with the previous criterion defined at lower altitude (900 hPa).

The advantages of this new criterion are:

- 700 hPa corresponds to ~3 km where most of the locations of the planet are lower than this altitude. It was the concern of how to use 900 hPa in high altitude regions (from a received personal comment).
- The results of using 700 hPa and -10 ºC or 900 hPa and 5 ºC do not change much qualitatively and quantitatively.
- The use of 700 hPa is more consistent with Saito et al. (2012), Montanyà et al. (2007) and Warner et al. (2014) for three different areas: Japan, Europe and US, respectively.
- The use of -10ºC is also convenient since might be related to the lower altitude of the main negative charge and it is also related to the non-inductive charging mechanisms.

**page 6, Chapter 4 and Fig. 5: Tw should be "the number of winter thunderstorm days per year" and not the number of thunderstorms.**
Agreed, thanks so much.

**Anonymous Referee #2**

**The paper "Global Distribution of Winter Lightning: a threat to wind turbine and aircraft" provides new analysis of global lightning data, focusing on the distribution of winter thunderstorms, that are known to be threat to aviation and wind turbines. The data analysis approach is novel, as provides a useful comparison of the relative intensity of regions of winter thunderstorms. The quality of the analysis and report suggests that it is of a sufficient standard to be published without major corrections; however the number of minor corrections required is significant. A very high proportion of these are due to language issues, where the point made is acceptable, but the wording requires clarification. Additionally, there are also a number of scientific questions that need to be addressed.**

We sincerely thank so much to Referee #2 for the exhaustive and constructive revision of the manuscript. We appreciate it.

**- GENERAL COMMENTS**

**The use of a threshold of 5 C at 900hPa is not sufficiently explained/justified. Why isn't 4 C, or 6 C, appropriate? What is the impact of using different temperature/pressure levels? It may be that this approach has been justified in existing literature, but if this is the case, an appropriate reference would be required.**

Same response as to Referee #1:

It is difficult to establish a precise criterion and easy enough to allow the global classification of winter lightning and thunderstorms. Our criterion of temperatures equal or lower than 5 ºC at 900 hPa produce good agreement with the observations indicated in the referenced papers. But we decided to modify the criterion to (700 hPa and -10ºC ) in the way to be more consistent with the references of Saito et al. (2012), Montanyà et al. (2007) and Warner et al. (2014). We choose actually, our criterion of winter lightning to be those lightning flashes occurring in cold airmass thunderstorms. Then a criterion of -10 ºC at 700 hPa is more reasonable in order to identify airmass thunderstorms and not include those storms such as winter squall lines (common in the US) which are associated with warm and stable air above the freezing air. The results obtained are very similar with the previous criterion defined at lower altitude (900 hPa).

The advantages of this new criterion are:

- 700 hPa corresponds to ~3 km where most of the locations of the planet are lower than this altitude. It was the concern of how to use 900 hPa in high altitude regions (from a received personal comment).
- The results of using 700 hPa and -10 ºC or 900 hPa and 5 ºC do not change much qualitatively and quantitatively.
- The use of 700 hPa is more consistent with Saito et al. (2012), Montanyà et al. (2007) and Warner et al. (2014) for three different areas: Japan, Europe and US, respectively.
- The use of -10ºC is also convenient since might be related to the lower altitude of the main negative charge and it is also related to the non-inductive charging mechanisms.

**The use of these specific criteria also means that thunderstorms at high latitudes in the summer are still classified as "winter" thunderstorms. This can be seen in Figure 4, where lightning in the meteorological summer is classified as winter lightning in North America and northern Asia (bottom left subfigure) and near to the Southern Atlantic and Southern Indian Ocean (top right subfigure). The term is appropriate in general, but this characteristic should be discussed, as it may be that these (technically) summer thunderstorms actually exhibit characteristics similar to winter thunderstorms (i.e. in terms of the heights of charged regions).**

Our criteria for winter thunderstorms here is not straightly related to the year's winter season. We do not filter by season. Here our winter thunderstorms are those occurring in cold air masses and as pointed by the reviewer, certainly storms in high latitudes can have this type of thunderstorms in winter. We have clarified that in the introduction and at the beginning of section 3.2 where we introduce the methodology for obtaining the maps.

**Although the comparison of the relative intensity of global regions of winter lightning activity is useful, the authors do not provide enough analysis of the global variability in WWLLN detection efficiency (DE).In page 6, line 31, they state that the DE of WWLLN "is considered to be 11%". There will be significant variations in this globally, however, based on sensor distribution. Further to this, Rudlosky and Shea (2013) showed that WWLLN is apparently three times more likely to detect a flash over the ocean than over land, based on comparison with LIS data. (Rudlosky, Scott D., and Dustin T. Shea. "Evaluating WWLLN performance relative to TRMM/LIS." Geophysical Research Letters 40, no. 10 (2013): 2344-2348)**

Thanks for the comment. That is an interesting point. WWLLN data provides relative hourly DE and that might be used. Actually, we considered that but two concerns aroused: first the DE is relative, not absolute. Second, if it would be realistic for those winter thunderstorms that produce very few flashes. If i.e. a location with 0.05 relative DE has 1 stroke a simple approach would be to compensate and assume actually 20 strokes. That might be very unrealistic. Areas with low relative DE (e.g 0.05) can have strong influence. We improved the text addressing the reader to the provided references and others that discuss and provide maps of relative DE of WWLLN and the reader can do the judgment for any possible correction with DE.

By the way, this comment is also very interesting in relation to the use of LLS data for risk assessment. Standards such as IEC61400-24 proposes a risk assessment in order to estimate the number of lightning flashes to a wind turbine. There are interesting discussions in the group of IEC related to with standard in order how to compensate DE of LLS data and also how to deal with winter thunderstorms (e.g. March 2015, APL conference).

**The figures present the results in units of strokes per square kilometer per year. Is there a reason that WWLLN fixes were not merged into flashes using suitable space/time criteria? This would make the data easier to compare with satellite data (OTD, LIS, or the upcoming GLM and MTG-LI) and other analyses of lightning density, which also generally use flashes.**

Thanks for the comment. One reason why we do not group WWLLN detections in to flashes is because the computation would take much longer. The second is that we are not sure about grouping with WWLLN since it uses TOGA technique. Grouping strokes into flashes also requires some good location accuracy if the common criteria of 1 s maximum duration, 500 ms of maximum inter-stroke interval and 10 km between strokes.

**The middle paragraph of the Discussion section (Page 7, lines 12-21) seems a little muddled. The authors leap from a one-sentence analysis of one area, to another area, to another. The Mediterranean and New Zealand are mentioned before North America and the North Atlantic, despite the latter experiencing much higher winter lightning densities than the former, based on Figure 3. More context in the text, i.e. peak densities or winter thunderstorms days, would make this section simpler to understand without constantly needing to refer to the figures.**

**The colour scales used in Figures 4 and 5 could be adjusted to allow for the appreciation of greater detail in regions of stroke densities above 0.2 strokes per square kilometer per year. Large regions of North America, North Atlantic and the North Pacific are simply displayed as block red in Figure 3. The current colour scale between 0.0 and 0.2 could be kept the same, so that 0.2 is still red, but values about that could transition into black/grey/purple, for example. See Anderson and Klugmann (2014) as an example.**

We improved the figures using a non-rainbow stile color bar. We substituted the old 'jet' colorbar (rainbow type) for a non-rainbow type (e.g. Matlab 'parula') which can provide a linear representation.
(for more information see: http://www.alecjacobson.com/weblog/?tag=colormap).

**There are a large number of specific comments and technical corrections that are simply due to the use of language, but do not prohibit understanding of the topics discussed. References were generally accessible and appropriate for the text, unless otherwise stated in the specific comments.**

**SPECIFIC COMMENTS**

**Page 1, line 9: "for new generation of aircraft." does not read well in English.**
**Suggest "for modern aircraft."**
Ok, thanks.

**Page 1, line 10: "That is because the use of lightweight of" does not read well in English. Suggest merging with the previous sentence: "for modern aircraft, due to the use of lightweight composite materials."**
Done, thanks so much.

**Page 1, line 14: "characterized for producing" does not read well in English. Suggest "characterized by a relatively high proportion of"**
Done, thanks.

**Page 1, line 14: "type" Should be "types"**
Done, we appreciate your very detailed review.

**Page 1, line 15: "However, up to now**
**" The however is unnecessary, and "up to"does not read well. Suggest "Until now"**
Done, many thanks.

**Page 1, line 17: "three maps" Actually, six maps are presented, in three figures.**
Done, thanks so much.

**Page 1, line 21: "provided here can be used for providing" Double use of forms of the word "provide" does not read well. Further, the information presented in the manuscript is not sufficient for an entire assessment of risk on its own. Suggest "provided here can be used in the development of a"**
Done, we appreciate your very detailed review.

**Page 1, line 26: ""But, although" Word "But" is not necessary.**
Done, many thanks.

**Page 1, line 26: "... winter thunderstorms is low" It would read better is this were put in context. Suggest "winter thunderstorms is relatively low compared with summer thunderstorms"**
Done, thanks.

**Page 2, line 9: "events to airplanes" does not read well in English. Suggest "events involving airplanes" or "events involving lightning connections to airplanes"**
Done, thanks so much.

**Page 2, line 12: "This phenomena present" Mixes singular and plural, should read" This phenomena presents"**
Agreed.

**Page 2, line 13: "under this conditions." Mixes singular and plural, should read "under these conditions."**
Thanks.

**Page 2, line 15: "interactions highlighting the situations related" does not read well in English. Suggest "interactions related"**
Thanks so much again.

**Page 2, line 15: "That is" does not read well in English. Suggest "This is"**
Agreed.

**Page 2, line 21: "can receive both downward and initiate upward" The word "both" refers to receiving downward and initiating upward lightning, and so should appear earlier. Suggest "can both receive downward and initiate upward**

Done, we appreciate your very detailed review.

**Page 2, line 23: "We do that" does not read well in English. Suggest "We do this"**

Agreed.

**Page 2, line 31: Fig. 1d is referred to in the text before Figs 1b and 1c. It would make sense then to either rearrange this sentence, so Fig. 1d is mentioned after the previous figures, or to rearrange the subfigures in Figure 1 so that what is currently 1d becomes 1b, 1b becomes 1c and 1c becomes 1d.**

Fixed.

**Page 3, line 1: It has been contested whether IC lightning is technically a "flash" in the same way as CG lightning, as there is not a stepped leader/return stroke structure. I have debated this previously with other scientists, who prefer the term "IC discharge". The change here is not an absolute necessity, however.**

That is an interesting comment. We like more to use the term flash since lightning flashes can produce or not a ground stroke. With the lightning mapping array (e.g one of our references such as van der Velde and Montanyà, 2013) one can see the complexity of lightning flashes. There, the leaders to ground appear as another characteristic of a flash. We use the term lightning flash for all types of lightning and when there is a contact to ground we name it as CG flash and when there is no contact to ground simple IC flash to the entire complex lightning leader activity.

**Page 3, line 1: "Upward induced lightning is more likely to occur" It would be useful to clarify whether this refers to absolute numbers of upward induced lightning events or relative numbers. As an arbitrary example, if the relative proportion of events drops by 50%, but the total amount of lightning increases by a factor of 4, the absolute number doubles.**

That is a very interesting question and difficult for us to provide quantitative numbers. Observations such as Tom Warner in South Dakota towers most of the upward events are induced by close lightning flashes. We observed similar pattern in Europe for summer and some self-initiated (measurements at our instrumented tower) during winter thunderstorms.

**Page 3, line 16: "because of the necessity of" This does not read well in English. Suggest " because the presence of"**

Thanks so much.

**Page 3, line 16: "charge region necessary to" This does not read well in English. Suggest "charge region is necessary to"**

Agreed.

**Page 3, line 29:** "turbines belongs to" Mixes singular and plural, should read "turbines belong to"
Thanks so much again.

**Page 3, line 30:** "raised the interest." This does not read well in English. Suggest" has been discussed and investigated."
Agreed.

**Page 4, line 1:** "an hour especially when" Suggest inserting a comma, i.e. "an hour, especially when"
Thanks.

**Page 8, line 1:** "But Japan is" This does not read well in English. Suggest "Japan is"
Agreed.

**Page 8, line 2:** "lightning but other" This does not read well in English. Suggest" lightning, as"
Thanks so much again.

**Page 8, line 4:** "The maps are helpful for risk assessment" This has not been conclusively demonstrated yet. Suggest "The maps may be of use for risk assessment"
Agreed, thanks so much again.

**Page 8, line 4:** "IEC". Expand this acronym.
Done.

**Page 8, line 5:** "turbines can be submitted to" This does not read well in English. Suggest " turbines can be exposed to"
Agreed, thanks.

**Page 8, line 6** "for self-lightning initiation." This does not read well in English. Suggest "for lightning self-initiation."
Done, thanks.

**Page 8, line 8:** "Also high risk locations are those offshore" This does not read well in English. Suggest "Locations at the greatest risk tend to be offshore"
Agreed, thanks.

**Page 8, line 9:** "the new presence of" This does not read well in English. Suggest" the installation of "
Thanks, agreed.

**Page 8, line 11:** "winter lightning will allow to conduct further." This does not read well in English. Suggest "winter lightning may be beneficial in conducting"

Agreed, thanks so much.

**Page 8, line 12: "on local lightning data and meteorological data." This does not read well in English. Suggest " on local combined lightning and meteorological data."**

Agreed, many thanks.

**Page 9, lines 21-23: Montanyà el al. (2011) reference not generally available online.**

This is a conference reference that we can provide under request.

**Page 9, line 29: Murooka (1992) reference not accessible unless via web page in Japanese.**

Agree, a summary of the observations of Murooka (1992) can be found in other papers and books (e.g. Lightning protection of aircraft, by Franklin A. Fisher, J. Anderson Plumer, Rodney A. Perala).

**Page 10, line 21-22: Wang and Takagi (2011) not generally available online.**

ICAE conference papers can be easily obtained by request to the secretary of the ICAE: http://icae.jp/

**Figure 1: Subfigures 1a and 1b imply that the CG lightning is initiated from the small positively charged region at the base of the cloud. Assuming these figures represent negative CG lightning, it would be better if the branching of the lightning were to extend further into the negatively charged region, and were to spread out within that region (Montanya et al. 2014a, Figure 3 demonstrates the extent of a CG flash within a cloud nicely).**

The sketch has been adapted according the reviewer's suggestion. The initiation is between the positively and negatively charged regions and we included some branches extended into the negatively charged region.

**Figure 1, caption: Remove "(Case of Japan)", this does not make sense.**

This text is clarified. It is substituted by: "charge distribution as observed winter storms in Japan" since the plotted electric charge structure corresponds to the winter thunderstorms in Japan which are characterized by the production of positive CG flashes. That might be not the same case for other regions like in Europe where most of the lightning flashes in winter storms are negative.

**Figure 1, caption: "Proportions are not meet in the representations." This does not read well in English. Suggest "Proportions in these diagrams are not to scale."**

Done, thanks.

**Figures 3-5: The font within the figures is slightly too small, and difficult to read. The coastlines and edges of the plots are also very fine. It may be that the figures were created at one size, then had to be scaled down to fit the page. It would be preferable to scale down the original image, so that when it is reproduced in print, the text and edge lines appear less fine.**

The figures have been improved according to this comment.

**Figure 4: I have a suspicion that the values of the flash densities in this figure are wrong. If flash densities across North America/Japan are well in excess of 0.2 flashes per square kilometer per year annually, and there are seasons where the values are well below this level, there must be seasons where the flash density is significantly greater than the annual average. Looking at the peak densities in Figure 4, this is not the case. What I suspect has happened is that the number of flashes per grid box have only been divided by the area, to give units of flashes per square kilometer per season. In fact they must then be divided by the number of days in the year and multiplied be the number of days in the season to convert the units to flashes per square kilometer per year. This means that all of the densities are too low by a factor of four.**

Thanks for the comment. The values in Fig. 4 are calculated for each grid box as the number of strokes divided by the area and averaged for the 5 year period. In such case, for a particular grid box, the addition of the four values sums the density in Fig 3. That is clarified in the text. We see the reviewer's point and agree that we should take into account for calculating the density the period of three months instead and entire year. However, we prefer to keep as we calculated in order to keep the addition of the four periods as the total annual density. We have indicated and clarified now this in the text and in the caption.

**Figure 4: Subfigure titles contain "900 05": Presumably these are the temperature height and cut-off settings for the data. These would preferably be removed.**

These labels have been removed. Thanks.

**Figure 4: The details in the maps are hard to see in this plot. There are two options that would improve this situation. One would be to use a single large, narrow colorbar to the right of all four figures, as the same scale is used in each, and the amount of whitespace could also be reduced, to make better use of the available space. Alternatively, the maps could be rotated by 90 degrees to fill a page sideways, which would allow for a lot more detail to be visible.**

The figures have been improved according to this comment. The text fonts are enlarged. As indicated before, we substituted the 'jet' colorbar (rainbow type) for a non-rainbow type (e.g. Matlab 'parula') which can provide a linear representation   ( for more information see: http://www.alecjacobson.com/weblog/?tag=colormap).

**- TECHNICAL CORRECTIONS**

We sincerely appreciate the very detailed review and the time dedicated to improve our paper. We have adopted all the following suggestions.

- **Page 1, line 24: "et al.,2015" should include a space after of comma, i.e. "et al., 2015".**
- **Page 1, line 24: "Anderson and D. Klugmann" should not include initial, i.e. "Anderson and Klugmann".**
- **Page 1, line 25: "Poleman" should be spelt "Poelman".**
- **Page 1, line 29: "(Montanyà et al., 2014)". There are two Montanyà et al papers in**
- **2014, this reference presumably refers to Montanyà et al., 2014a?**
- **Page 1, line 29: "Honjo, 2014". Wrong year, should read "Honjo, 2015".**
- **Page 2, line 10: "Wilkinson et al., (2009) concluded" Comma should be removed.**
- **Page 2, line 10: "Wilkinson et al., (2009) concluded" Incorrect year: should be 2013.**
- **Page 3, line 31: Missing caron and acute from name of author: Radicevic should read "Radicevic".**
- **Page 3, line 33: "Recently Montanyà et al. (2008) "No such reference: suggest Montanyà et al. (2014a).**
- **Page 3, line 34: "Fig. 1 f". Space between figure number and subfigure letter should be removed.**
- **Page 4, line 4: "Fig. 1b" should refer to "Fig. 1d".**
- **Page 4, line 8: "Fig. e and d" should refer to "Fig. 1e and 1f".**
- **Page 4, lines 9-10: "Montanyà et al. (2014)". There are two Montanyà et al papers in 2014, this reference presumably refers to Montanyà et al. (2014a)?**
- **Page 4, line 11: "Lightning" misspelled as "Lighnting".**
- **Page 5, line 16: "mid winter" should be "mid-winter".**
- **Page 5, line 25: "were" should be "where".**
- **Page 5, line 27: "artic" should be "Arctic".**
- **Page 5, line 27: "Holle and Watson, 1992". Wrong year, should read "Holle and Watson, 1996".**
- **Page 7, line 1: "Fig. 2 and 3" should be "Fig. 3 and 4".**
- **Page 7, line 25: Missing closing parenthesis.**
- **Page 8, line 19: Page numbers not needed in Abarca et al. (2010) reference.**

- **Page 8, line 22: Reference requires full title of paper: "A European lightning density analysis using 5 years of ATDnet data"**
- **Page 8, line 23: Incorrect year in Anderson and Klugmann (2013) reference: should be 2014.**
- **Page 8, line 26: Missing "and" between second to last and last author.**
- **Page 8, line 33: Incorrect year in Holle and Watson (1992) reference: should be 1996.**
- **Page 9, line 6: Page numbers not needed in Hutchins et al. (2012) reference.**
- **Page 9, line 25: Missing space in journal title: "Res.Atmos." should read "Res. Atmos."**
- **Page 9, line 30: Reference requires full title of paper: "The European lightning location system EUCLID – Part 2: Observations"**
- **Page 10, line 1: Missing caron and acute from name of author: Radicevic should read"Radicevic".**
- **Page 10, line 1: Missing space after colon: "Badea, I:Impact" should be "Badea, I:Impact"**
- **Page 10, line 10: Incorrect page range: "2653-2673" should be "2653-2674".**
- **Page 10, line 29: Missing space and full stop in author's name: "Williams, E.R," should be "Williams, E.R.,"**
- **Page 10, line 34: Missing indent at start of line.**
- **Figure 2: Caption reads "artic", should read "Antarctic".**

[revised manuscript text omitted]

5  The map in Fig. 3 clearly shows the previously discussed areas of winter lightning (Japan, east of US, Mediterranean) and other areas with wind farms such as Uruguay and surroundings, southwest of the Indian Ocean and Tasmanian Sea . Fig. 4 plots the seasonal variation of the global winter lightning activity. In this case, each grid cell corresponds to the five-year average value of the number of strokes divided by the cell area for the corresponding period. .

10  The average stroke densities shown in the maps in Fig. 3 and 4 are influenced by the detection efficiency of the WWLLN. As any long range VLF lightning location system, WWLLN has a detection efficiency for each location that changes during the hours of the day due to VLF wave propagation, and also during time due to network upgrades, sensitivity of the sensors and data processing methods. The estimated overall stroke detection efficiency of WWLLN is considered to be 11 % according to Hutchins et al., 2012, Abarca et al. (2010) and Rodger et al (2009). Although the relative detection efficiency is provided

15  periodically, we did not apply any compensation. For further information relative to the detection efficiency, see Hutchins et al. (2012). Moreover, the WWLLN makes no distinction between cloud-to-ground flashes and intra-cloud flashes. Then, the results presented in Fig. 3 and 4 are relative to WWLLN detections and cannot be adopted as absolute values.

Probably the most useful metric for evaluation of the winter lightning risk is the average number of winter thunderstorm days per year ($T_w$). Here $T_w$ is obtained in a 1°×1° grid. A winter thunderstorm day in a grid cell is counted if at least one lightning

20  stroke agreeing with the presented temperature-pressure level criterion is detected within the cell. The $T_w$ map is depicted in Fig. 5.

**4 Discussion**

Areas of winter lightning are well defined outside the ITCZ. The resulting maps show that contrary to what occurs with deep convective storms, winter lightning activity is more distributed over the oceans than over continental areas. One of the reasons

25  is the role that warm oceanic water plays to produce energy for convection during cold seasons. In Fig. 2 we resumed the main oceanic current gyres. Note how this simple picture goes a long way in explaining the patterns of winter lightning in Fig. 3 and 5. The preference for oceanic and coastal areas is an important aspect for coastal onshore and offshore wind farms.

Regarding winter thunderstorm activity and risk assessment, the maps indicate some particular areas more critical to winter lightning because of prevalence in larger areas onshore in proximity to active ocean areas. The Japan mainland is surrounded

30  by winter thunderstorms, the map in Fig. 5 shows how especially the west coast is particularly active reaching in some areas about  24 days of winter thunderstorms per year. Another particularly active region that affects extensive onshore areas is found in the Mediterranean Sea (e.g. South Italy and the Balkans). In the case of North America, the highest number of annual winter thunderstorm days per year is located in the Atlantic Ocean. Although the central eastern

regions of the US (Great Lakes) the number of winter thunderstorms per year are not so high, the lightning stroke densities are significant. Other areas sensitive to winter thunderstorms because present installations and also ongoing offshore farms, range from the northern coast of Spain, western coast of France, and the European coast in the North Sea. In addition, Uruguay and its surroundings, southern New Zealand, southern west coast of Chile and Southeast Alaska must be highlighted as well.

For risk assessment it is convenient to consider the $T_w$ provided in Fig. 5 as the first indicator of risk to winter lightning activity. In addition to identification of winter thunderstorm areas, the seasonal variation of the winter lighting activity (Fig. 4) is another aspect deserving attention for planning purposes, for equipment maintenance or to setup lightning warnings (e.g. European standard EN 50536 2011). That is also important when working with tall cranes. In the northern hemisphere the activity is concentrated from October to June whereas in the southern hemisphere the activity if significant from April to September.

**5 Conclusions**

Winter lightning poses a critical risk to tall objects such as wind turbines and also to flying aircraft because these have favourable conditions for self-lightning initiation. In this paper we presented for the first time world maps with winter lightning activity. The maps shows that winter lightning occurs in extratropical regions with preference for oceanic and coastal areas in the western limits of permanent oceanic gyres. As a general conclusion, winter lightning maps presented in this work suggest that winter activity in Japan may be the highest, as supported by the previously discussed works. Japan is not an exclusive region for winter lightning, as other areas such as the Mediterranean and the US are active as well. In the southern hemisphere,  Uruguay and its surroundings, the southwestern Indian Ocean and the Tasman Sea experience the highest activity.

The maps  may be of use for  risk assessment analysis such as proposed in standards (e.g. the International Electrotechnical Commission, IEC) providing a tool to identify areas of winter lightning activity. In these areas, tall structures such as wind turbines can be exposed to very energetic lightning and to an environment favourable for lighnting self  initiation. Risk assessment of the effect of winter lightning shall also include the exposure. In the case of wind turbines, those turbines located in areas influenced by winter lightning at high altitudes can experience very high number of lightning flashes. Locations at the greatest risk tend to be  offshore (e.g. offshore wind turbines, platforms and helicopter operations at those sites). In these situations, the  installation of a tall object can significantly increase the winter lightning activity.

Finally, the simple methodology employed to classify a lightning stroke as being winter lightning may be beneficial in conducting  further risk assessment based on local combined lightning  and meteorological data.

[revised manuscript text omitted]

10 Figure 4. Seasonal variation of the winter lightning stroke density  distribution for the period of 2009-2013. The values are calculated as the average number of strokes for the five years in each grid cell divided by the area of the cell. Note that major shift of activity from DJF and MAM periods in the northern hemisphere to more JJA and SON periods in the southern hemisphere.

[Figure]

Figure 5. Average number of winter thunderstorm days per year ($T_w$) for the period 2009-2013.